# Development and genetics of red coloration in the zebrafish relative *Danio albolineatus*

**Delai Huang[1], Victor M Lewis[1†], Tarah N Foster[2], Matthew B Toomey[2,3], Joseph C Corbo[3], David M Parichy[1,4]\***

[1]Department of Biology, University of Virginia, Charlottesville, United States; [2]Department of Biological Science, University of Tulsa, Tulsa, United States; [3]Department of Pathology and Immunology, Washington University School of Medicine, St Louis, United States; [4]Department of Cell Biology, University of Virginia, Charlottesville, United States

**Abstract** Animal pigment patterns play important roles in behavior and, in many species, red coloration serves as an honest signal of individual quality in mate choice. Among *Danio* fishes, some species develop erythrophores, pigment cells that contain red ketocarotenoids, whereas other species, like zebrafish (*D. rerio*) only have yellow xanthophores. Here, we use pearl danio (*D. albolineatus*) to assess the developmental origin of erythrophores and their mechanisms of differentiation. We show that erythrophores in the fin of *D. albolineatus* share a common progenitor with xanthophores and maintain plasticity in cell fate even after differentiation. We further identify the predominant ketocarotenoids that confer red coloration to erythrophores and use reverse genetics to pinpoint genes required for the differentiation and maintenance of these cells. Our analyses are a first step toward defining the mechanisms underlying the development of erythrophore-mediated red coloration in *Danio* and reveal striking parallels with the mechanism of red coloration in birds.

**\*For correspondence:**
dparichy@virginia.edu

**Present address:** †Institute of Molecular Biology, University of Oregon, Eugene, United States

**Competing interest:** The authors declare that no competing interests exist.

## Introduction

Red and orange pigments deposited in the skin provide key signals that are subject to sexual or natural selection. For example, the intensity of red or orange coloration, or the area over which it occurs, has been associated with mating preferences in a variety of species (*Milinski and Bakker, 1990*; *Hill, 1991*; *Houde, 1997*; *Grether, 2000*; *Takahashi, 2018*; *Ansai et al., 2021*) and in some cases the very conspicuousness that makes an individual attractive can make it more vulnerable to predators (*Endler, 1980*; *Godin and McDonough, 2003*; *Johnson and Candolin, 2017*). Red or orange coloration can also be associated with aggressive interactions (*Evans and Norris, 1996*; *Pryke and Griffith, 2006*; *Dijkstra et al., 2008*) and warning coloration (*Brodie and Brodie, 1980*; *Stevens and Ruxton, 2012*).

Red, orange, and yellow coloration is often mediated by the accumulation of carotenoids, fat-soluble compounds that are synthesized by plants and some fungi and bacteria, and obtained by animals via their diet and then subsequently modified (*Bagnara and Matsumoto, 2006*; *McGraw, 2006*; *Svensson and Wong, 2011*; *Sefc et al., 2014*; *Strange, 2016*; *Toews et al., 2017*). More than a thousand naturally occurring carotenoids are known, and their basic structure consists of 40 carbon atoms derived from four terpene molecules (*Figure 1A*). Carotenoids contain a system of conjugated double bonds that absorb light in the visible range; the greater the length of the conjugated system, the more red-shifted the absorption. Yellow carotenoids, such as β-carotene or zeaxanthin, have a total of 11 conjugated double bonds, whereas red carotenoids, such as astaxanthin, have a total of 13 due to the addition of ketone groups in the 4 and 4' positions of the terminal rings of the molecule

DOI: https://doi.org/10.7554/eLife.70253

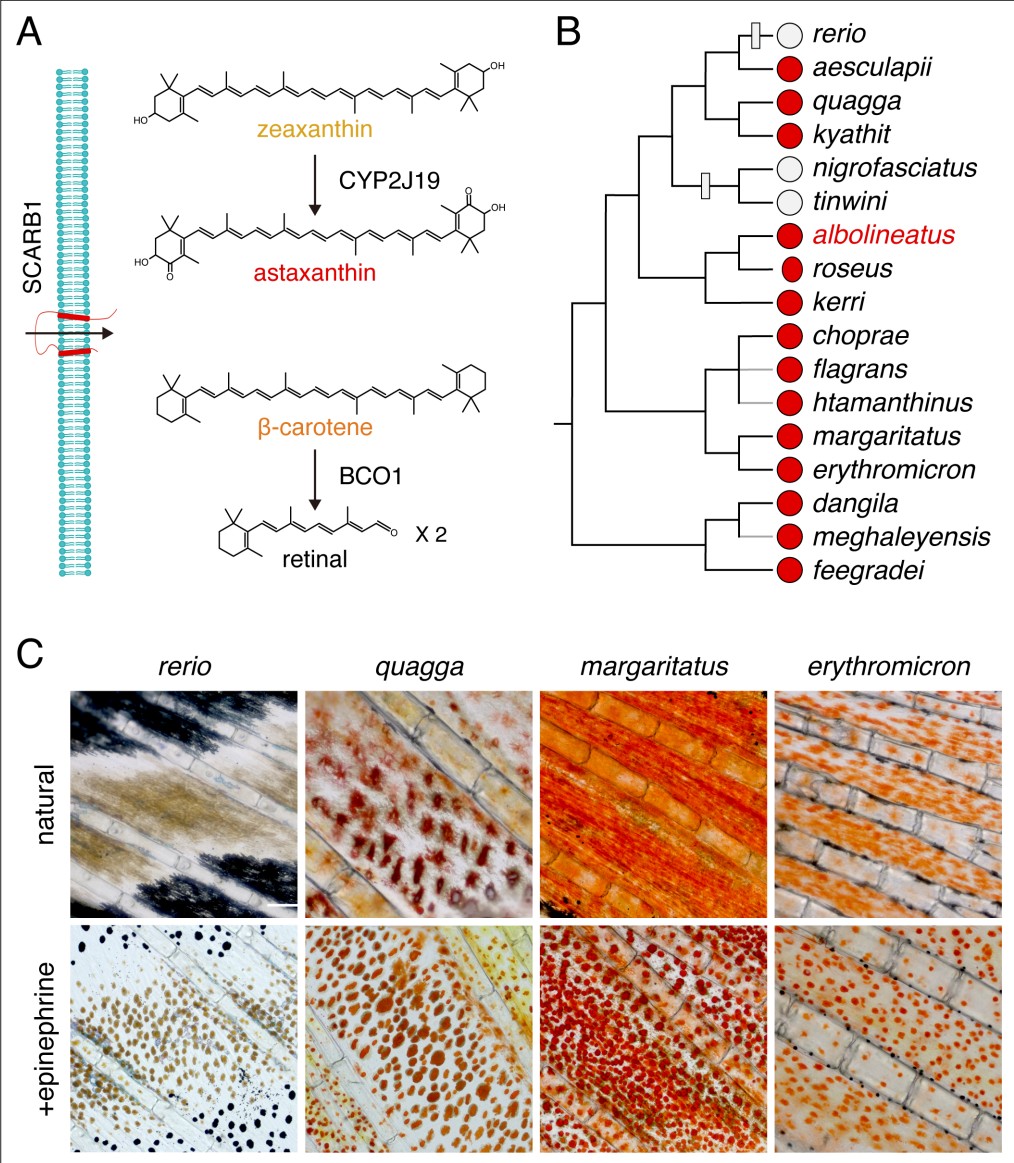

**Figure 1.** Carotenoid types and distribution of red erythrophores among *Danio* species. (**A**) Examples of major carotenoid types including yellow zeaxanthin, red astaxanthin, and orange β-carotene, with factors required for entry into cells and chemical modification (Main text). (**B**) Erythrophore presence (red circles) or absence (light gray circles) indicated by direct observation or prior species descriptions (*Fang and Kottelat, 2000*; *Quigley et al., 2005*; *Engeszer et al., 2007*; *Kullander and Fang, 2009*; *Kullander, 2012*; *Kullander and Norén, 2016*; *Spiewak et al., 2018*; *McCluskey et al., 2021*). A composite phylogeny based on several molecular evolutionary studies is shown; gray branches indicate lineage placements inferred by morphology alone (*Tang et al., 2010*; *Kullander, 2012*; *Kullander et al., 2015*; *McCluskey and Postlethwait, 2015*). Gray boxes across branches indicate lineages in which erythrophores are inferred most parsimoniously to have been lost. (**C**) Anal fin details of zebrafish (*rerio*) without erythrophores and other species with erythrophores. Cells are shown in their typical native states, with pigment dispersed, and following treatment with epinephrine, which causes pigment to be contracted toward cell centers. Scale bar: 100 μm.

---

(*Figure 1A*). Many animal species possess endogenous 'ketolase' activity, and thus can add ketones to diet-derived yellow carotenoids to produce red ketocarotenoids. The final color displayed by an animal reflects the specific carotenoids absorbed in the gut, transported to peripheral tissues, and chemically modified to achieve specific light-absorptive properties. Carotenoids also have important functions as vitamin A precursors, as antioxidants, and as regulators of a variety of cellular functions. The multifunctionality of carotenoids, and the need to obtain them from the environment, have

contributed to the notion that carotenoids in the integument can serve as an honest signal of prospective fitness, to mates or rivals (*Endler, 1980*; *Weaver et al., 2017*; *Weaver et al., 2018*).

The cellular context for displaying carotenoid-dependent colors differs between endothermic and ectothermic vertebrates. In birds, carotenoids are concentrated in keratinocytes of the skin and displayed either directly or after incorporation into feathers (*McGraw, 2006*). In ectotherms, carotenoids are concentrated in lipid droplets within pigment cells, chromatophores, visible through the epidermis and dermis. Red chromatophores are known as erythrophores, whereas yellow or orange chromatophores are referred to as xanthophores. Besides accumulating carotenoids, both cell types can produce and retain pteridine pigments that sometimes also contribute to visible coloration (*Schartl et al., 2016*; *Parichy, 2021*).

Erythrophores and xanthophores, like other skin chromatophores—black melanophores, iridescent iridophores, and white leucophores—develop from embryonic neural crest cells, either directly, or indirectly, via latent progenitors in the peripheral nervous system or other tissue compartments (*Kelsh et al., 2009*; *Patterson and Parichy, 2019*). Though sharing a common cell lineage overall, the degree to which different chromatophore types share fate-restricted progenitors remains incompletely understood. In zebrafish for example, most xanthophores on the body differentiate as pteridine-containing xanthophores in the embryo, then proliferate and lose their color, only to reacquire carotenoid-dependent coloration in the adult; other xanthophores on the body and in the fin develop instead from latent progenitors (*Tu and Johnson, 2011*; *McMenamin et al., 2014*; *Singh et al., 2016*; *Saunders et al., 2019*). By contrast, the majority of adult iridophores and melanophores develop from a common progenitor in the peripheral nervous system (*Budi et al., 2011*; *Dooley et al., 2013*; *Singh et al., 2016*). Lineage relationships can be further complicated by direct transitions between chromatophore types. Such transitions have long been known to be inducible experimentally (*Niu, 1954*; *Ide and Hama, 1976*), but have also been found naturally in zebrafish, in which a class of fin leucophore develops directly from melanophores (*Lewis et al., 2019*).

With respect to red coloration, the lineage relationship of erythrophores to xanthophores is not known. Although studies across several species have revealed both similarities and differences in cytological appearance and pigment biochemistry (*Matsumoto, 1965*; *Matsumoto and Obika, 1968*; *Ichikawa et al., 1998*; *Khoo et al., 2012*; *Djurdjevič et al., 2015*), it remains unclear if erythrophores share a progenitor with xanthophores, or whether they might develop directly from xanthophores. Likewise, the genes required for red coloration, as opposed to orange or yellow coloration, remain largely unexplored, although two loci required for, or associated with, ketocarotenoid accumulation, have been identified in birds (*Lopes et al., 2016*; *Mundy et al., 2016*; *Toomey et al., 2018*).

Here, we exploit the presence of erythrophores in a zebrafish relative, the pearl danio *Danio albolineatus*, to interrogate cell lineage relationships between erythrophores and xanthophores, and to identify genes essential for red and orange coloration in this species. By clonal analysis and fate mapping we show that early erythrophores and xanthophores of the fin arise from a common, initially orange progenitor, the descendants of which adopt one or the other fate depending on their location. We further show that later-arising erythrophores and xanthophores of the fin develop directly from unpigmented precursors, and that transitions between erythrophore and xanthophore states can occur during regeneration. By screening candidate genes identified through transcriptomic comparisons of erythrophore- and xanthophore-containing fin tissues, we additionally demonstrate requirements for several genes in red or yellow coloration. These include loci encoding a cytochrome P450 monooxygenase, belonging to the same protein family as an enzyme previously implicated in avian red coloration (*Lopes et al., 2016*), as well as two genes not previously implicated in red coloration. These results lay the groundwork for future biochemical analyses of carotenoid processing, dissection of mechanisms of erythrophore fate specification, and comparative analyses of species-specific losses or gains of erythrophore-dependent coloration.

## Results

### Phylogenetic distribution of erythrophores in *Danio* and patterning of erythrophores and xanthophores in the anal fin of *D. Albolineatus*

The adult zebrafish (*Danio rerio*) pigment pattern includes yellow xanthophores, black melanophores, at least three types of iridescent iridophores, and two types of white cells (melanoleucophores and

xantholeucophores) (*Hirata et al., 2003*; *Lewis et al., 2019*; *Patterson and Parichy, 2019*; *Gur et al., 2020*). Zebrafish does not have red erythrophores. Because erythrophores occur in many other species of teleosts, we surveyed the distribution of these cells across the *Danio* genus more broadly. Of 17 *Danio* species assessed, 14 had erythrophores indicating this cell type is common and most likely was present in the common ancestor of all *Danio* species (*Figure 1B–D*).

We focused on *D. albolineatus* since erythrophores are abundant in this species and are separated spatially from other pigment cells, an arrangement likely to facilitate analysis (*Goodrich and Greene, 1959*). In the anal fin of adults, red erythrophores were located proximally and were separated from the more distal yellow xanthophores by a narrow stripe of melanophores (*Figure 2A*). Although erythrophores were present in both sexes, the cells were more deeply and consistently red in males than females and we therefore focused on males at stages when sexes were distinguishable (*Figure 2—figure supplement 1A*). Male fish older than 1 year often lacked fin stripe melanophores, indicating that some pattern remodeling continues even after sexual maturation (*Figure 2—figure supplement 1B*). As compared to xanthophores, erythrophores occur at lower densities and were more likely to be binucleated (*Figure 2B*, middle and right panels; *Figure 2—figure supplement 1C and D*), a characteristic associated with a mature state of differentiation in stripe melanophores of zebrafish (*Saunders et al., 2019*).

Because subtle differences in color can be difficult to discern, we sought metrics to describe mature and developing cells, under brightfield and fluorescent illumination in which these cells were distinguishable as well: xanthophores displayed green autofluorescence upon excitation with blue-green light (488 nm) owing to the presence of yellow carotenoids (*Granneman et al., 2017*; *Saunders et al., 2019*), whereas erythrophores showed only weak autofluorescence at this wavelength but much stronger red autofluorescence upon exposure to green-yellow light (561 nm)(*Figure 2C*). We then compared hue in brightfield illumination with relative red:green autofluorescence, confirming the differences between erythrophores and xanthophores (*Figure 2D and E*).

To understand the anatomical context of erythrophore development, we imaged fish during the larva-to-adult transformation. The first pigmented cells in the anal fin were lightly melanized melanophores. Subsequently, orange xanthophore-like cells appear that were pale and had smaller areas of visible pigment than mature erythrophores and xanthophores (*Figure 2B*, left panel; *Figure 2—figure supplement 2*). These early orange cells autofluoresced in both red and green channels, consistent with their appearance in brightfield (*Figure 2F*). During later development, however, cells in proximal regions were increasingly red whereas cells in distal regions were increasingly yellow. Cell densities gradually diverged between proximal and distal regions as well (*Figure 2F*; *Figure 2—figure supplement 1D*).

## Fin erythrophores and xanthophores arise from a common progenitor

As a first step in dissecting lineage relationships of erythrophores and xanthophores, we sought to determine whether these cells arise from a common early progenitor. Since red and yellow colors were likely carotenoid-based, we reasoned that lineage relationships should be revealed by clones of cells in fish mosaic for *scavenger receptor b1* (*scarb1*), which is essential for carotenoid accumulation in avian integument and zebrafish xanthophores (*Toews et al., 2017*; *Toomey et al., 2017*; *Saunders et al., 2019*). If erythrophores and xanthophores share a lineage, then rare wild-type clones should contain both red and yellow cells in an otherwise colorless background. If erythrophores and xanthophores have distinct lineage origins, however, wild-type clones should often contain only red cells or only yellow cells. In *D. albolineatus* injected with high efficiency AltR CRISPR/Cas9 reagents targeting *scarb1*, wild-type clones most often contained both red cells and yellow cells (*Figure 3A*).

We further assessed relationships by labeling individual clonal lineages by *tol2* transgenesis. We found that orange cells of larvae and both erythrophores and xanthophores of adults expressed transgenes driven by regulatory elements of *aldehyde oxidase 5* (*aox5*) isolated from zebrafish. *aox5* functions in the synthesis of pteridines present in xanthophores and erythrophores (see below) and is expressed by xanthophores and their specified precursors in zebrafish (*Parichy et al., 2000*; *McMenamin et al., 2014*). When we injected an *aox5* reporter transgene at limiting dilutions to express membrane-targeted EGFP, labeled cells were restricted to narrow regions along the anterior–posterior axis, consistent with derivation from single clones observed in other contexts (*Tu and Johnson, 2010*; *Singh et al., 2014*; *Spiewak et al., 2018*). Such cells occurred on the body and fin,

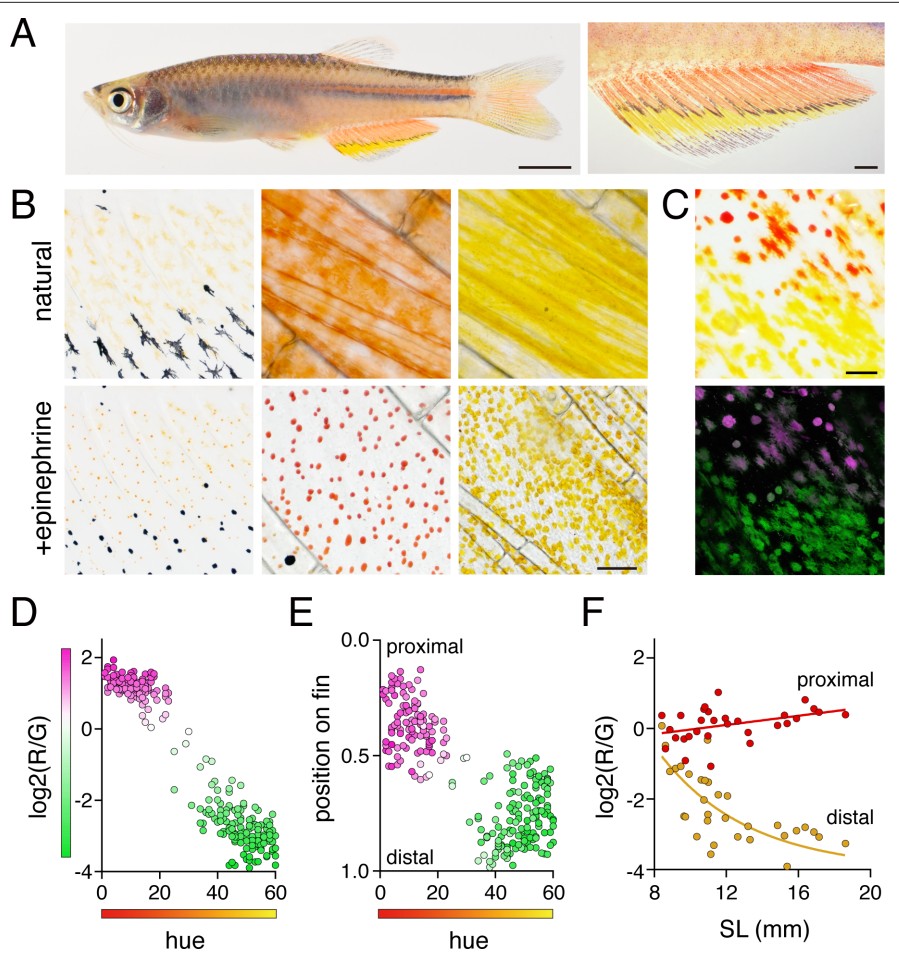

**Figure 2.** Anal fin pigment pattern of *D. albolineatus* and its ontogeny. (**A**) Erythrophores were present on the body and were particularly evident on the anal fin (closeup at right), where these cells were found more proximally than yellow xanthophores. (**B**) At larval stages xanthophore-like cells with a uniform orange coloration occurred across the entire fin (left panels). Later in the adult, proximal red erythrophores and distal yellow xanthophores have distinct colors (middle and right panels). (**C**) Erythrophores and xanthophores had different spectra under epiflourescence. Erythrophores autofluoresced in red (displayed in magenta) whereas xanthophores autofluoresced in green. (**D**) Hue values under brightfield illumination were correlated with ratios of red to green autofluorescence ($R^2 = 0.92$, p < 0.0001). (**E**) Colors of cells varied across the proximodistal axis of the fin, shown as relative position with fin base at 0 and fin tip at 1. Erythrophores in proximal regions were distinct in both fluorescence ratio and visible hue from xanthophores in distal regions though some intergradation was evident in middle regions, near the melanophore stripe. N = 250 cells from five adult males in D and E. Color fills represent red to green fluorescence ratios. (**F**) During the larva-to-adult transformation, ratios of red to green autofluorescence diverged between prospective erythrophore and xanthophore regions. Individual red and yellow points correspond to mean values of cells in in proximal and distal regions, respectively, from each of 31 male or female fish (N = 620 cells total) imaged at a range of developmental stages represented by different standard lengths (SL). Scale bars: 5 mm (**A**, left), 1 mm (**A**, right); 25 µm (**B, C**).

The online version of this article includes the following figure supplement(s) for figure 2:

**Figure supplement 1.** Sex and age differences in erythrophore pigmentation and occurrence of a binucleated state.

**Figure supplement 2.** Pattern development in anal fins of larva to juvenile fish.

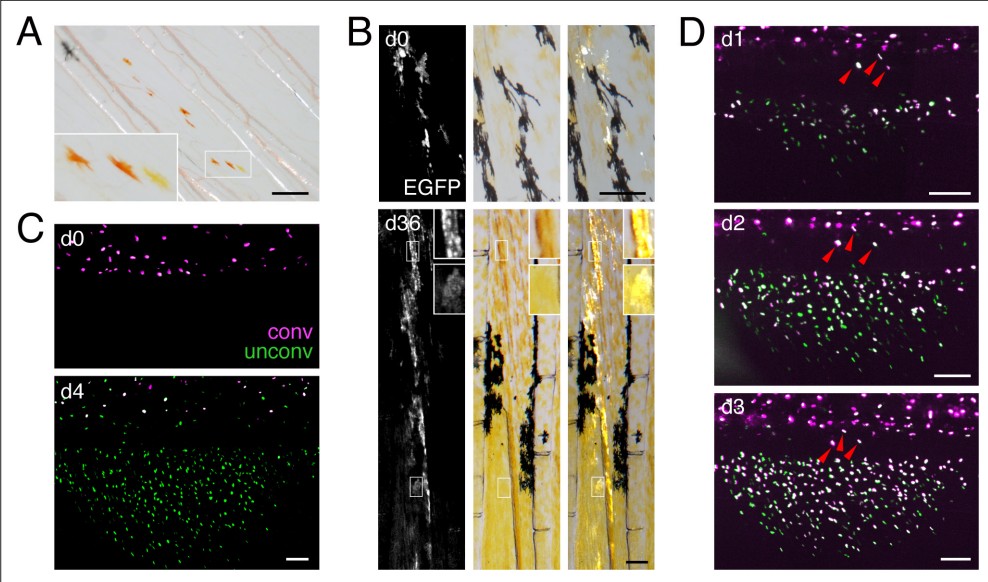

**Figure 3.** Shared progenitor of fin erythrophores and xanthophores revealed by clonal analyses. (**A**) In fish mosaic for somatically induced mutations in *scarb1* most rare, wild-type clones consisted of both erythrophores and xanthophores (8 of 10 presumptive clones in seven fish, with remaining clones only containing one or the other cell type; an additional 56 fish derived from injected embryos either lacked wild-type cells or lacked mutant cells and were thus uninformative). (**B**) Clonal labeling of xanthophores and erythrophores with *aox5:palmEGFP*, illustrating flourescence, brightfield, and merged views of the same fields. In the clone shown here, an initial complement of several orange cells at the level of the melanophore stripe (d0, 7.5 mm SL) expanded to include more cells proximally and distally to the melanophore stripe that differentiated as erythrophores and xanthophores, respectively (d36, 15 mm SL; red arrowheads). For these analyses, limiting dilutions of *aox5:palmEGFP* were injected into ~500 embryos, yielding 271 embryos that exhibited some fluorescence at 3 days post-fertilization that were further sorted at 16 dpf, identifying 27 individuals with patches of expression in the anal fin. Of these 27 fish, one subsequently died and eight were found to have broad expression across the entire fin, likely representing multiple clones of uncertain boundaries, and so were excluded from analysis. The remaining 18 fish exhibited 24 spatially distinct, presumptive clones of *aox5:palmEGFP*-labeled cells, of which 22 presumptive clones contained both erythrophores and xanthophores as shown here [consistent with mixed clones of melanophores and xanthophores in zebrafish (***Tu and Johnson, 2010***; ***Tu and Johnson, 2011***)]; one clone contained only erythrophores and one clone contained only xanthophores. (**C**) When *aox5:nucEosFP+* cells on the body were bulk photoconverted before fin development, only unconverted *aox5*:nucEosFP+ cells (green nuclei) were present in the fin 4 days later (images representative of all N = 3 fish tested). (**D**) Successive steps in anal fin development and erythrophore/xanthophore lineage specification revealed many cells newly acquiring *aox5:nucEosFP* expression at daily intervals within the fin (green nuclei). Though some *aox5:nucEosFP+* cells were present at the fin base these did not enter into the fin proper (white cells, arrowheads; images shown are from a single individual representative of all N = 7 fish tested in this manner over 23 days each). Scale bars: 200 µm (**A, B**); 100 µm (**C, D**).

The online version of this article includes the following figure supplement(s) for figure 3:

**Figure supplement 1.** Transgene labeling of erythrophores and xanthophores.

and in these presumptive clones, erythrophores and xanthophores were almost always co-labeled (***Figure 3B***; ***Figure 3—figure supplement 1***), consistent with a common progenitor for erythrophores and xanthophores.

A common progenitor could be specified for erythrophore or xanthophore fates either before or after colonizing the fin. To distinguish between these possibilities we used *aox5* reporter expression as an indicator of specification and a nuclear localizing photoconvertible (green→red) fluorophore, nucEosFP, to determine whether cells already expressing this marker transit from body to fin. We generated a transgenic line, Tg(*aox5:nucEosFP*)*vp37albTg*, which allowed us to photoconvert all *aox5:nucEosFP+* cells on the body prior to anal fin development (6.5 mm SL). We then assessed the distribution of converted and unconverted nucEosFP 4 days later, when the anal fin had started to form (7.5 mm SL). Because *aox5* expression persists once initiated, cells photoconverted at one stage will later have converted fluorophore (displayed in magenta), as well as new, unconverted fluorophore

(green), so nuclei will appear white; cells that initiate *aox5* expression only after photoconversion will have only unconverted fluorophore and nuclei that are green.

We found that cells on the body had converted and unconverted fluorophore, whereas cells in the fin had only unconverted fluorophore, consistent with initiation of *aox5* expression only after progenitors had colonized the fin (*Figure 3C*). Because it remained possible that some cells had migrated from body to fin and proliferated so extensively that signal of converted nucEosFP was lost by dilution, we repeated these analyses but assessed distributions of cells 1 d after photoconversion; we then photoconverted (or reconverted) all cells on body and fin and repeated this process on successive days. Such labeling failed to reveal cells that translocated from body to fin, although it did reveal numerous cells that acquired *aox5* expression when already in the fin (*Figure 3D*). Together these observations suggest that progenitors migrate to the fin, become specified for erythrophore or xanthophore lineages within the fin, and then contribute to both populations as they proliferate to populate the proximal–distal axis during fin outgrowth.

## Erythrophores and xanthophores arise from fate-restricted and unrestricted precursors in the fin and their fates remain plastic even after differentiation

Clones identified by *scarb1* activity or *aox5* transgene expression (*Figure 3A and B*) likely represented progenitors segregated from other lineages during early development when injected Cas9 is active and transgene integration occurs (e.g., *Tryon et al., 2011*); these clones presumably also included melanophores or other cell types not revealed by these markers (*Tu and Johnson, 2010*; *Tu and Johnson, 2011*; *Lewis et al., 2019*). We therefore asked whether progeny of such clones that had already colonized the fin were restricted to either erythrophore or xanthophore fates by photoconverting individual nucEosFP+ cells at early stages of fin development (7.0–8.5 mm SL) and then assessing phenotypes of resulting clones 30–36 days later (15.0 mm SL). At 7.0 mm SL, only unpigmented nucEosFP+ cells were present (*Figure 4A*, top). Preliminary observations indicated that proximally located cells tended to remain in the proximal region where erythrophores develop, so we photoconverted cells in distal regions that might become more broadly distributed. Resulting clones consisted of erythrophores if daughter cells remained relatively proximal as the fin grew out, or both erythrophores and xanthophores if daughter cells became distributed across the proximodistal axis (*Figure 4B and C*; *Figure 4—figure supplement 1A*). At 7.5 mm, many nucEosFP+ cells had acquired a pale orange color (*Figure 4A*, bottom) and so we asked whether these cells had become fate-restricted with the onset of pigmentation. Similar to unpigmented cells, however, initially proximal orange cells generated only erythrophores, whereas initially distal orange cells could generate clones of only erythrophores, both erythrophores and xanthophores, or only xanthophores, depending on where daughter cells were distributed (*Figure 4C*; *Figure 4—figure supplement 1B*). Finally, at 8.5 mm we found that still-unpigmented nucEosFP+ cells near the distal fin tip generated distal clones restricted to a xanthophore fate (*Figure 4—figure supplement 1C*). These results show that individual unpigmented cells and early-developing orange cells in the fin can generate both erythrophores and xanthophores, depending on initial location and where progeny localize.

We further asked whether phenotypes of erythrophores and xanthophores might be plastic even after they differentiate by challenging cells in a regenerative context. To test for erythrophore → xanthophore conversion, we amputated fins through the region containing erythrophores, expecting that regeneration distally might allow for repositioning of erythrophores into regions where regenerative xanthophores would be expected, with conditions favorable to fate conversion, should cells retain such potential. We first assessed the possibility that transfating occurs by repeatedly imaging individual fish in brightfield, to learn whether cells near the amputation plane might lose their red color during regenerate outgrowth. Individual erythrophores could often be reidentified using other cells as well as distinctive features of fin ray bones and joints as landmarks (*Figure 5A*; *Figure 5—figure supplement 1*). As regeneration proceeded, small groups of cells having paler red or orange coloration, were sometimes observable where individual cells of deep red coloration had been found, suggestive of proliferation and dilution of pre-existing pigments. Later, only yellow cells were found in these same locations. These observations were consistent with the possibility of erythrophore → xanthophore conversion. To test this idea directly we marked nucEosFP+ erythrophores by photoconversion prior to amputation and followed labeled cells through regeneration (*Figure 5B*; *Figure 5—figure*

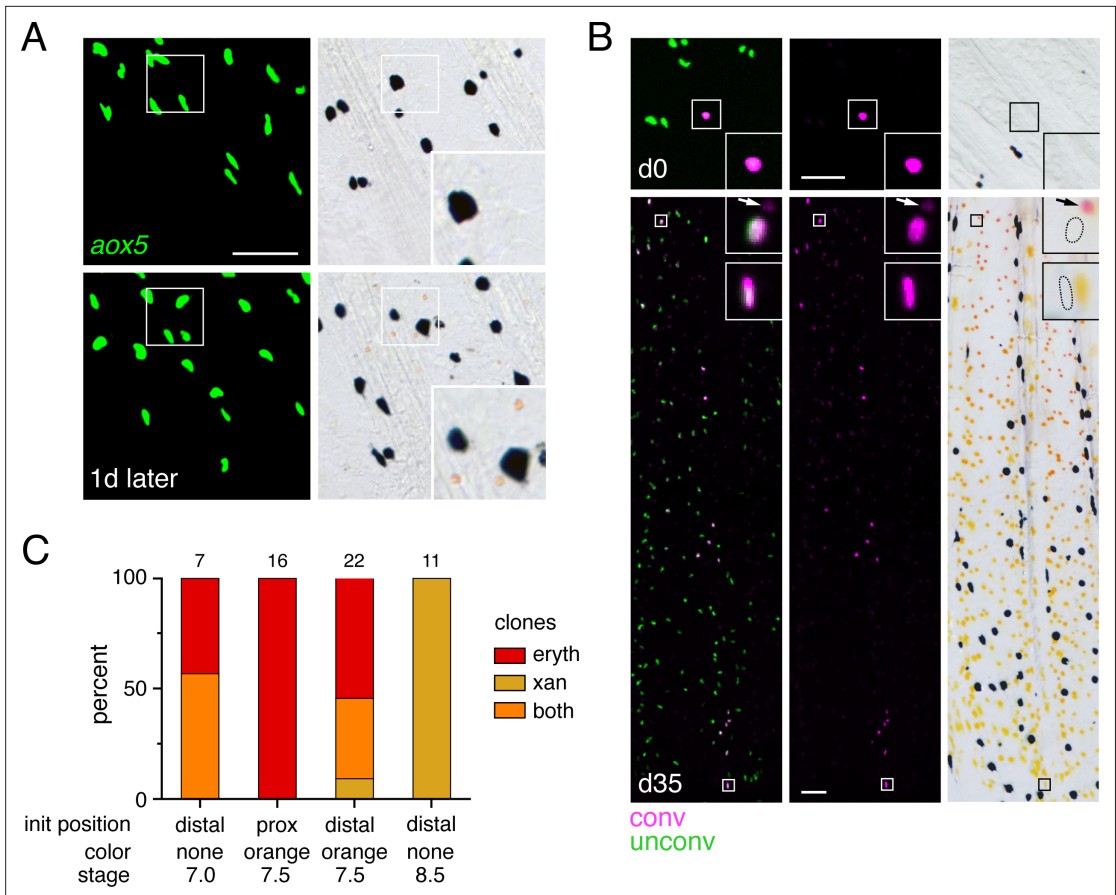

**Figure 4.** Bipotential precursor to erythrophores and xanthophores revealed in the fin by fate mapping. (**A**) Unpigmented cells of the xanthophore lineage, marked by *aox5:nucEosFP* transgene expression (see Main text), present at 7.0 mm SL had acquired a pale orange color 1 day later. (Representative of all N = 7 fish examined by repeated imaging during larval development.) Insets show higher magnification images of a corresponding region. (**B**) Example of a photoconverted, initially unpigmented cell (d0, 7.0 mm SL) that yielded a clone containing both erythrophores and xanthophores (d35, 15.0 mm SL; representative of four of seven clones, with remaining clones containing erythrophores only). Fish were treated with epinephrine to contract pigment before imaging. Arrows indicate erythrophore autofluorescence from red carotenoid pigment, which accumulates adjacent to nuclei following epinephrine treatment; approximate positions of nucEosFP+ nuclei in brightfield images are shown with dashed outlines. Insets, proximal and distal cells in the clone. (**C**) Percentages of clones containing only erythrophores, only xanthophores, or both cell types. Numbers above bars indicate clone sample sizes examined. In these analyses pigment cells and progenitors stably expressed *aox5:nucEosFP* (7.5, 8.5 mm SL) or mosaically expressed a different transgene, *mitfa:nucEosFP* (7.0 mm SL), that had been injected into embryos at the one-cell stage. In zebrafish, *mitfa* (*melanophore-inducing transcription factor a*) is expressed by pigment cell progenitors, as well as melanophores and xanthophores (**Lister et al., 1999**; **Budi et al., 2011**; **Saunders et al., 2019**), and we found in *D. albolineatus* that *mitfa:nucEosFP* was expressed in these cells as well as orange cells of larvae and erythrophores of adults. *mitfa:nucEosFP* was used for fate mapping at early stages owing to its more robust expression in unpigmented cells. Scale bar: 50 μm.

The online version of this article includes the following figure supplement(s) for figure 4:

**Figure supplement 1.** Fate mapping of single photoconverted cells at different locations and stages.

*supplement 2A*). Many erythrophores divided to replenish their complement in proximally regenerating tissue, and a few erythrophores differentiated from unpigmented precursors, as indicated by the presence, or absence of photoconverted nucEosFP, respectively (*Figure 5—figure supplement 2B*). Additionally, some initially marked erythrophores came to occupy more distal regions and were indistinguishable from regenerative xanthophores that had developed from unpigmented progenitors even 36–51 days post-amputation (*Figure 5C*). These findings suggest a reduction in pre-existing red pigment as cells divide, and a failure to accumulate new red pigments once proliferation has ceased. We also sought to determine whether xanthophores can transition to an erythrophore fate by ablating central regions of fin and then assessing whether distal xanthophores can move into the regenerating proximal region. However, these experiments were not informative, as regenerative tissue was

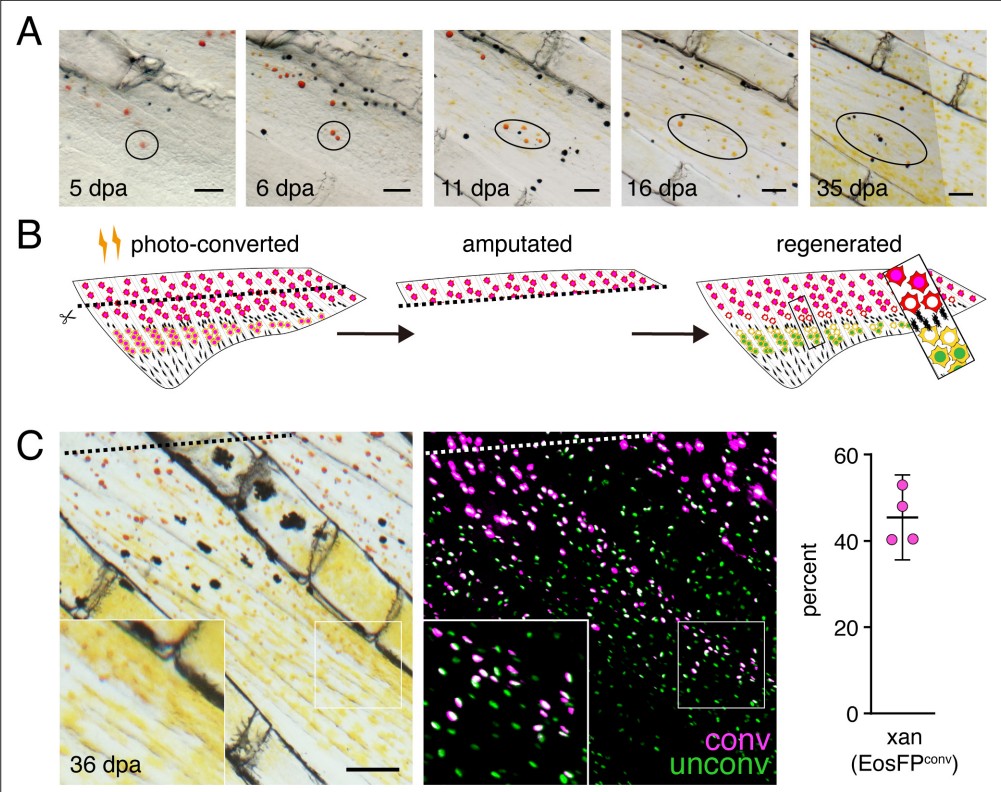

**Figure 5.** Regeneration assays reveal fate plasticity in differentiated cells and latent stem cells competent to differentiate as erythrophores and xanthophores. (**A**) Brightfield sequence of regeneration illustrating apparent conversion of erythophores to xanthophores (image series representative of all N = 3 fish examined by repeated imaging through regeneration). As fins regenerated, individual erythrophores (circled) near the amputation plane appeared to divide, with presumptive daughter cells having reduced amounts of pigment visible upon contraction with epinephrine and an increasingly yellow–orange color. (**B**) Schematic of regeneration experiment in C. Fins of Tg(*aox5:nucEosFP*) fish were photo-converted *in toto* prior to amputation through the erythrophore region. Fins regenerated over 15 days and pigment pattern had re-formed by 30 days, at which time a new melanophore stripe and distinct regions of erythrophores and xanthophores had developed. (**C**) Example of cells in regenerative tissue 36 days post-amputation (dpa). Regenerative xanthophores near the plane of amputation often contained photoconverted nucEosFP in a region of fin extending 400 µm from the distalmost red erythrophore into the regenerative xanthophore region (means ± 95 % confidence interval; N = 1964 cells in four fish examined). Dashed lines indicate amputation in B and C. Scale bars: 50 µm (**A**); 100 µm (**C**).

The online version of this article includes the following figure supplement(s) for figure 5:

**Figure supplement 1.** Pigment cell arrangements and colors and fin tissue context during pattern regeneration.

**Figure supplement 2.** Regeneration of erythrophores from newly specified unpigmented progenitors.

**Figure supplement 3.** Regeneration of central fin regions.

---

colonized by erythrophores or xanthophores differentiated from progenitors rather than pre-existing xanthophores (*Figure 5—figure supplement 3*).

These observations indicate that erythrophores and xanthophores of the adult anal fin share a lineage, that individual progenitor cells within the fin can contribute to both cell types, and that some plasticity in fate persists even after differentiation, with erythrophores able to transition to a yellow-pigmented phenotype when challenged to do so.

## Genetic requirements and biochemical basis for red coloration

To better understand molecular mechanisms of red coloration we compared gene expression between fin regions containing only erythrophores or only xanthophores. Erythrophores occurred at only ~one-third the density of xanthophores in two dimensional images (*Figure 2—figure supplement 1D* and see below) and proximal and distal fin regions presumably differ in ways other than chromatophore

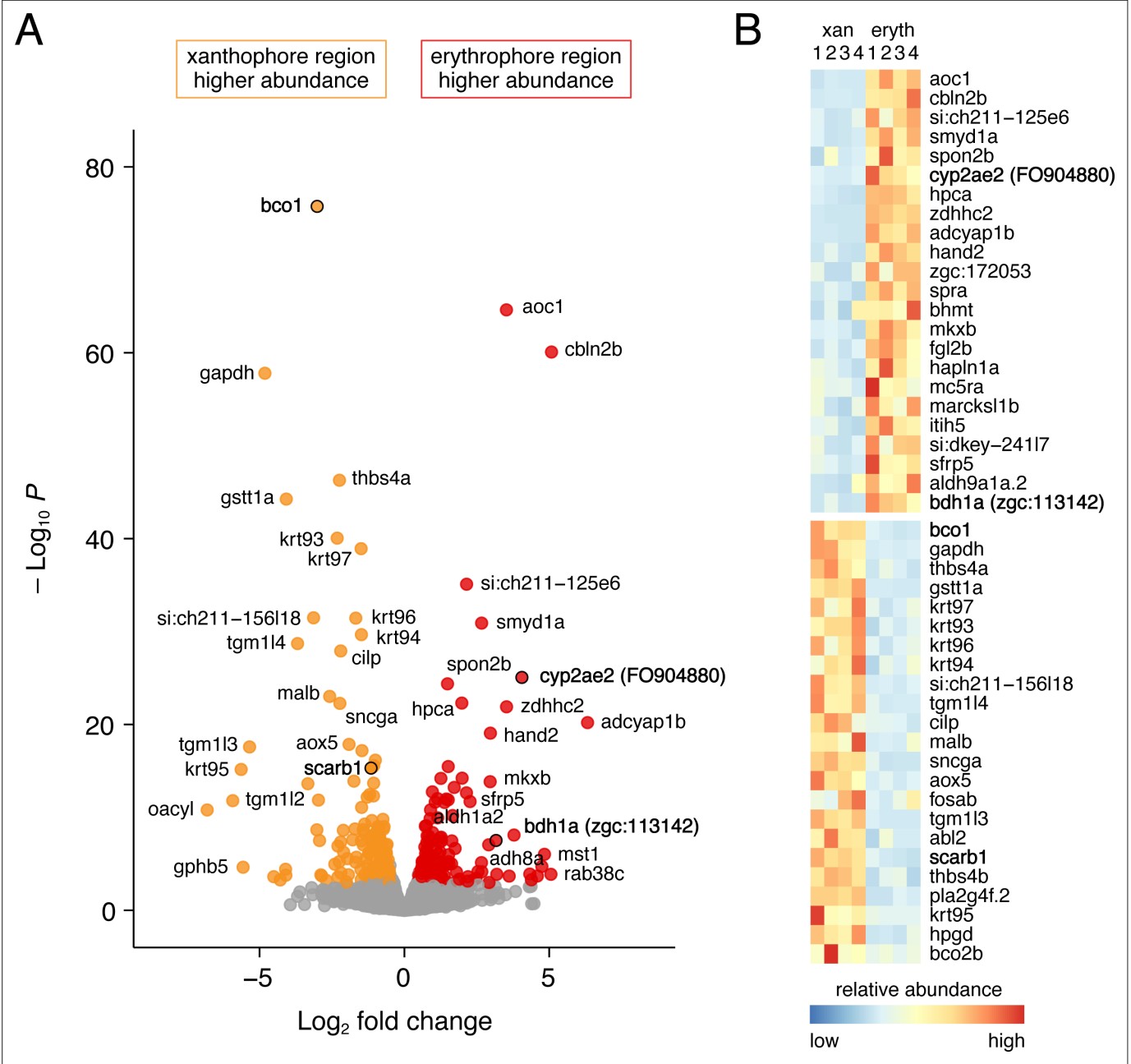

**Figure 6.** Differential gene expression in fin regions with erythrophores and xanthophores. (**A**) Volcano plot of detected transcripts. Yellow–orange and red points indicate transcripts more abundant in xanthophore-containing and erythrophore-containing regions, respectively (q ≤ 0.05). Gray points, transcripts not significantly different in abundance between regions. (**B**) Heat maps illustrating differential expression of selected loci across fin regions and replicate libraries. Genes with names in bold had phenotypes affecting erythrophore pigmentation.

content. Nevertheless, we reasoned that comparisons of bulk tissue preparations might still identify genes having marked differences in expression between erythrophores and xanthophores, as would be expected for loci functioning in pigment synthesis (*Saunders et al., 2019*). Mapping *D. albolineatus* sequencing reads to the zebrafish genome identified 18,050 expressed genes. Transcripts of 162 genes were more abundant in proximal erythrophore-containing tissue, whereas transcripts of 200 genes were more abundant in distal xanthophore-containing tissue (q < 0.05; fold-changes = 0.4–6.8) (*Figure 6A and B*; *Supplementary file 1*—Table 1).

To identify genes required for red or yellow coloration, we used CRISPR/Cas9 mutagenesis to knock out selected candidates that might have roles in processing of carotenoids, synthesis of other

pigments, or fate specification (***Supplementary file 1***—Tables 2 and 3). We screened mosaic (F0) fish and isolated stable lines of mutant alleles for target genes with pigmentary phenotypes. Of 25 targets derived from RNA-seq, three yielded mutants with defects in pigmentation. To determine which pigments contributed to colors present in wild-type, and which were affected in mutants, we further assayed the carotenoid content of fin regions by HPLC.

In the wild type, fin tissue containing erythrophores was markedly enriched for the red ketocarotenoid astaxanthin; additional peaks had profiles consistent with other ketocarotenoids (***Figure 7A***, peak 3; ***Figure 7—figure supplement 1***; ***Supplementary file 1***—Table 4). Fin tissue containing xanthophores lacked astaxanthin and instead contained yellow zeaxanthin (peak 10), similar to zebrafish xanthophores (***Saunders et al., 2019***), as well as additional peaks characteristic of other yellow xanthophyll carotenoids.

To confirm that carotenoids rather than other pigments are principally responsible for pigmentation, we recovered mutant alleles of *scarb1*, required for carotenoid uptake and localization (***Toomey et al., 2017***; ***Saunders et al., 2019***), as residual color in such mutants would suggest a non-carotenoid contribution. We isolated two alleles, *scarb1*$^{vp38ac1}$ (V84Δ16X) and *scarb1*$^{vp38ac2}$ (V84X), and found that *scarb1*$^{vp38ac1/vp38ac2}$ individuals had a phenotype concordant with that of F0 mosaics (***Figure 3A***): they lacked color in the visible range and lacked carotenoids detectable by HPLC (***Figure 7B***; ***Figure 7—figure supplement 2A***; ***Supplementary file 1***—Table 4). The absence of residual red or yellow coloration suggested that pteridine pigments do not contribute to visible color in these cells, as they do in some other species (***Goodrich et al., 1941***; ***Matsumoto and Obika, 1968***; ***Grether et al., 2001***; ***Weiss et al., 2012***; ***Olsson et al., 2013***). Moreover, targeting of differentially expressed genes known to function in pteridine synthesis did not yield visible pigmentation defects in F0 mosaics (erythrophore region: *spra*, *xdh*; xanthophore region: *aox5*; ***Supplementary file 1***—Tables 1 and 4). Pteridine pigments were detectable histologically in erythrophores and xanthophores, however, and could be visible to fish in the UV range (***Figure 7—figure supplement 2B***).

Mutants for two genes, *cyp2ae2* (*FO904880.1*) and *bdh1a* (*zgc:113142*), lacked overt red coloration (***Figure 7C***; ***Figure 7—figure supplement 2A***), although densities of erythrophores and xanthophores did not differ significantly from wild-type (***Figure 7F***). Both genes had transcripts that were more abundant in fin tissue containing erythrophores than xanthophores (log$_2$ fold-changes = 4.9, 2.1; q = 9.6E-23, 8.9E-8; ***Figure 6***). We confirmed by RT-PCR that both genes were expressed in erythrophores picked manually by micropipette from dissociated fin tissue (***Figure 7—figure supplement 3A***).

*cyp2ae2* encodes an enzyme within the large family of cytochrome P450 monooxygenases (***Kirischian et al., 2011***). A related gene encoding a different P450 family member, *CYP2J19*, is essential for red coloration in 'red factor' canary (***Lopes et al., 2016***) and zebra finch (***Mundy et al., 2016***), and likely has similar roles in other birds and turtles (***Twyman et al., 2016***; ***Twyman et al., 2018***). *CYP2J19* expression is testosterone-dependent (***Khalil et al., 2020***) and its product is believed to play an essential role in the conversion of yellow carotenoids like zeaxanthin into red ketocarotenoids like astaxanthin (***Figure 1A***). Orthologs of *CYP2J19* appear to be restricted to birds and turtles (***Twyman et al., 2016***). Reciprocally, *cyp2ae2* (*FO904880.1*) is clearly a member of the cyp2 family, and is likely orthologous to *cyp2ae1* loci of other teleosts by sequence similarity and chromosomal position, yet no clear orthologues of this gene are present in amniotes (Ensembl Release 103) (***Kirischian et al., 2011***; ***Yates et al., 2020***). The loss of red color in *D. albolineatus* erythrophores thus raises the possibility that CYP2J19 and Cyp2ae2 may have acquired carotenoid ketolase activity convergently. Supporting the idea that *cyp2ae2* might encode a carotenoid ketolase, the *cyp2ae2* mutant had markedly reduced amounts of astaxanthin in fin regions containing erythrophores but relatively greater amounts of zeaxanthin (peak 10) in both erythrophore and xanthophore containing tissue as compared to wild-type, consistent with *cyp2ae2* expression in both tissues, albeit at different levels (***Figure 7—figure supplement 1***, ***Supplementary file 1***—Table 5). Consistent with these findings, erythrophores of *cyp2ae2* mutants had markedly reduced red/green fluorescence ratios (***Figure 7D***) and a reduced diameter of visible pigment (***Figure 7E***). Trace residual astaxanthin and other ketocarotenoids are unlikely to reflect residual activity of *cyp2ae2*, as the mutant allele, *cyp2ae2*$^{vp39ac1}$, harbors a five-nucleotide frameshift within the first coding exon leading to 42 novel amino acids followed by a premature stop codon (L46Δ43X). A paralogous locus, *cyp2ae1*, lies adjacent to *cyp2ae2* and was expressed at very low levels in both fin regions (***Figure 7—figure supplement 3B***; ***Supplementary***

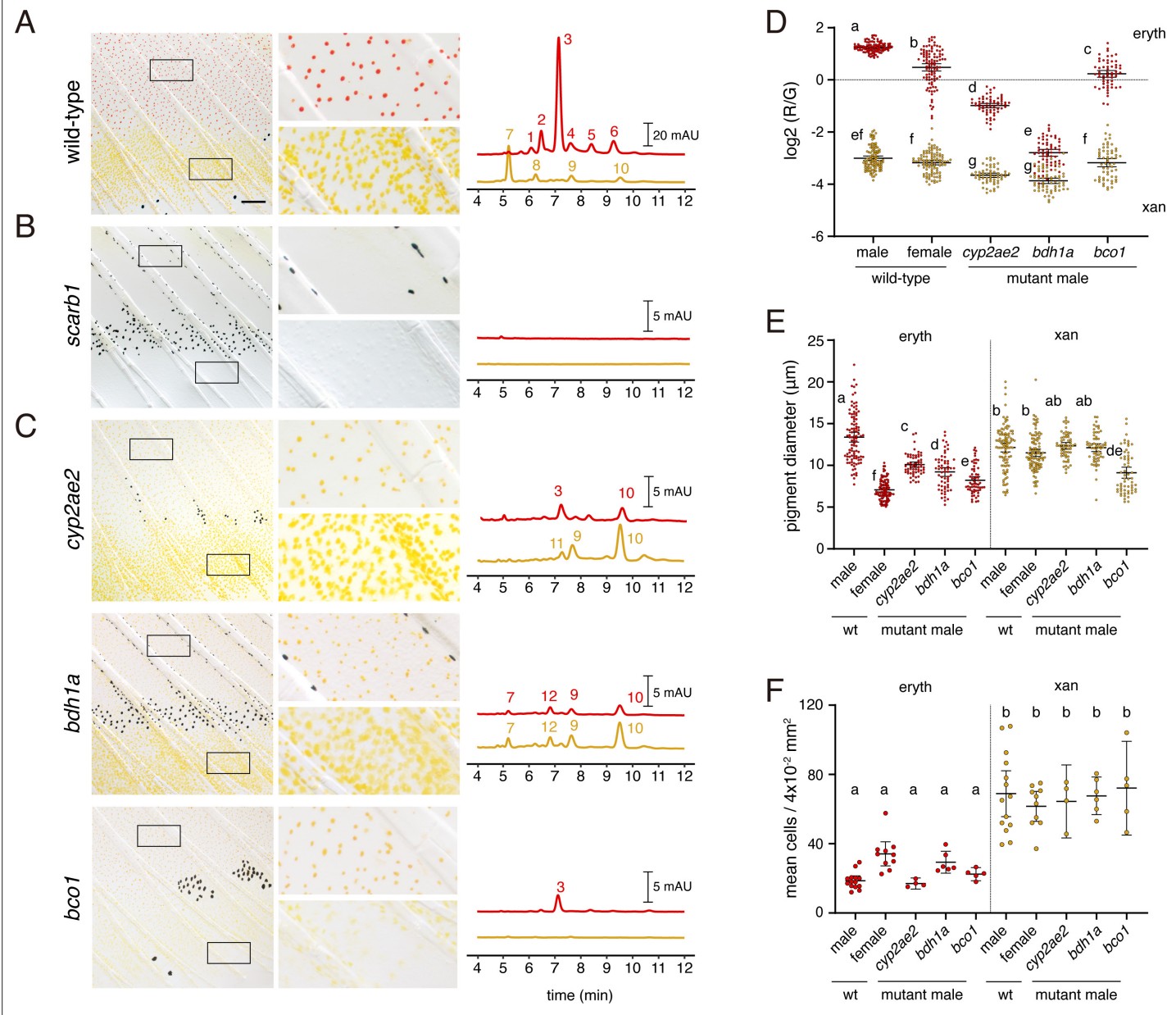

**Figure 7.** Wild-type pigment composition and mutant phenotypes. (**A**) Wild-type fin and carotenoid profile, showing carotenoid absorbance at 455 nm in adult male proximal tissue (red) and distal tissue (yellow). Numbers indicate different carotenoid species, with the most abundant ketocarotenoid in erythrophore-containing tissue being astaxanthin (peak 3; *Figure 1A*; *Figure 7—figure supplement 1*). (**B**) Homozygous *scarb1* mutants lacked red and yellow coloration and carotenoids were not detectable. (**C**) Homozygous mutant phenotypes of genes targeted from RNA-Seq comparisons. *cyp2ae2* and *bdh1a* mutants were deficient for red color and astaxanthin. *bco1* mutants had reduced red and yellow coloration and carotenoids. (**D**) Ratios of red to green autofluorescence for cells found within proximal erythrophore containing regions (red filled points) and distal xanthophore containing regions (yellow filled points) of wild-type males and females compared to mutant males. In the wild-type, erythrophores and xanthophores were segregated into different populations by R/G fluorescence, although differences in females were less marked. In males of each mutant, R/G ratios of erythrophores were reduced compared to wild-type, and lesser reductions were evident in xanthophores (ANOVA, genotype x region interaction, $F_{4,736}$=310.82, p < 0.0001, after controlling for significant main effects and variation among individuals; N = 760 cells total from five individuals of each background). Plots show means ±95 % confidence intervals; means of groups not sharing the same letter differed significantly from one another (p < 0.05) in Tukey-Kramer *post hoc* comparisons. (**E**) Wild-type males and females, and mutant males, differed in total visible pigment, as measured by diameters of contracted pigment granules following epinephrine treatment (*Saunders et al., 2019*). (ANOVA, background x region interaction, $F_{4,736}$=76.25, p < 0.0001, with significant main effects and variation among individuals; diameters were *ln*-transformed for analysis to control for increasing residual variance with means.). (**F**) Densities of erythrophores and xanthophores differed across backgrounds ( ANOVA, background x region interaction, $F_{1,35}$=19.01, p < 0.0001). Each point represents the mean number of cells counted in three regions of 4 × 10⁻² mm² in proximal or distal regions with erythrophores or

*Figure 7 continued on next page*

*Figure 7 continued*

xanthophores, respectively, in each of 39 total fish. Scale bar: 50 μm.

The online version of this article includes the following figure supplement(s) for figure 7:

**Figure supplement 1.** Characterics of carotenoid absorbance spectra.

**Figure supplement 2.** Mutant lesions recovered, presence of pteridines, and mosaic phenotype of *bco1*.

**Figure supplement 3.** Expression, genomic location and additional phenotypes of genes contributing to red coloration.

*file 1*—Table 1). Compensatory activity of *cyp2ae1* might account for trace levels of ketocarotenoids in *cyp2ae2*[vp39ac1].

The second red-deficient mutant, *bdh1a*, encodes 3-hydroxybutyrate dehydrogenase type 1a, a short-chain dehydrogenase/reductase. Homologues of this gene in mammals are known to interconvert hydroxyl and ketone groups, and in particular acetoacetate and 3-hydroxybutyrate, two major ketone bodies (*Green et al., 1996*; *Langston et al., 1996*; *Persson et al., 2009*; *Otsuka et al., 2020*). Although not implicated previously in carotenoid processing or red coloration, transcripts of a homologous gene were enriched in orange skin of clownfish *Amphiprion ocellaris* (*Salis et al., 2019*). The *bdh1a*[vp40ac1] mutant (D141Δ12X) completely lacked astaxanthin and other ketocarotenoids in erythrophore-containing tissue, and did not exhibit increased levels of zeaxanthin, as observed in *cyp2ae2*[vpa39c1] (*Figure 7C*). Red/green fluorescence ratios of erythrophores are similar to those of xanthophores (*Figure 7D*) and the diameter of visible pigment is reduced from wild-type levels in erythrophores though not xanthophores (*Figure 7E*), confirming the visible phenotype.

Erythrophores of *cyp2ae2* and *bdh1a* mutant fish appeared normal in size and shape in young adults yet became morphologically heterogeneous as fish age, with pigment-containing cell fragments and fewer cells evident, as well as an onset of whole-fish kyphosis by ~12 months post-fertilization (*Figure 7—figure supplement 3C*). These phenotypes suggest requirements for both loci in the accumulation of red carotenoids and subsequent homeostasis of erythrophores and other tissues, perhaps associated with a systemic dysregulation of carotenoid–Vitamin A—retinoid metabolism (*von Lintig et al., 2005*; *Ghyselinck and Duester, 2019*).

β-carotene oxygenase 1 (Bco1) symmetrically cleaves β-carotene to produce vitamin A, a precursor of retinoic acid; whereas β-carotene oxygenase 2 (Bco2) cleaves a variety of carotenoids in an asymmetric fashion, often leading to their degradation (*Widjaja-Adhi et al., 2015*; *Li et al., 2017*; *Harrison and Kopec, 2020*; *Poliakov et al., 2020*). *bco1* and *bco2b* were more abundant in tissue containing xanthophores than erythrophores (log$_2$FC = 4.9, 2.1; $q$ = 9.6E-23, 8.9E-8); a third locus, *bco2l* was similarly abundant at both sites (*Supplementary file 1—Table 1*). Given heterogeneities in transcript abundance, we asked whether β-carotene oxygenase genes might also contribute to differences in carotenoid accumulation between cell types. Only *bco1*-targeted fish exhibited an overt pigmentary phenotype in F0 mosaic animals, with pigment-free patches alongside patches of cells having apparently normal pigmentation consistent with a pigment-cell autonomous function (*Figure 7—figure supplement 2C*). Mutants stably carrying *bco1* alleles (*bco1*[vp41ac1], 52DΔ5X; *bco1*[vp41ac2], 51FΔ9X) had reduced carotenoid levels in both xanthophores and erythrophores as well as smaller diameters of contracted pigment granules (*Figure 7C–E*; *Figure 7—figure supplement 2A and B*). The mechanism of this effect on chromatophore carotenoid content remains unclear.

## Discussion

Red and orange coloration play important roles in multiple behaviors, including mate choice (see Introduction). As a first step toward understanding the development of such colors and the mechanisms underlying their phylogenetic distribution in the zebrafish genus *Danio*, we investigated the cell lineage origins and genetic requirements for erythrophore differentiation in *D. albolineatus*. These analyses provide new insights into the diversification of adult pigment cell types in teleosts and identify genes contributing to the red ketocarotenoid coloration in this species and possibly more distant taxa as well.

Fate mapping and clonal analyses indicated that at least some erythrophores and xanthophores share a common progenitor in the fin. Clones of cells marked genetically during early development later contained both cell types, indicating a shared progenitor that likely colonizes the fin during

its initial outgrowth, consistent with inferences for melanophore and xanthophore progenitors of zebrafish fins (*Tu and Johnson, 2010*; *Tu and Johnson, 2011*). In *D. albolineatus*, these progenitors appear to become specified for erythrophore or xanthophore fates—as inferred from *aox5* transgene expression—only after colonizing the fin. We did not observe pigmented cells transit from the body to the fin, though such cells could be found at the base of the fin without entering the fin itself. These findings might appear to differ from that of a prior study, in which erythrophores on the body were described as invading the fin (*Goodrich and Greene, 1959*). Yet those observations were made with the caveat that an appearance of invasion could also reflect de novo differentiation within regions not previously occupied by these cells, rather than active migration per se. We conclude that unpigmented progenitors enter the fin and only then become specified to erythrophore or xanthophores fates.

At early stages of fin outgrowth, some initially unpigmented progenitors acquire an orange color, intermediate between that of fully differentiated erythrophores and xanthophores. When these early orange cells were marked individually by photoconversion of a transgenic reporter, some cells initially at middle positions along the fin proximodistal axis gave rise to both erythrophores and xanthophores, whereas other more proximal or more distal cells contributed to only erythrophore or xanthophores, respectively. These observations indicate a bipotentiality, with fate choice presumably dependent on factors in the fin environment. Our finding that erythrophores can lose their red color when joining a regenerative population of xanthophores further indicates a subsequent plasticity in these fates. At later stages of fin development, unpigmented cells developed directly as xanthophores in distal regions. Whether these represent a distinct sublineage remains to be determined.

Red and orange colors can be generated in various ways. In mammals, reddish hues typically depend on the production of phaeomelanin by melanocytes, which is transferred to keratinocytes for incorporation into hair (*Slominski et al., 2005*; *Hubbard et al., 2010*; *Tadokoro and Takahashi, 2017*; *Caro and Mallarino, 2020*). In birds, phaeomelanin is also known to contribute to brownish-red feather coloration (*McGraw et al., 2005*; *Cruz-Miralles et al., 2020*), but more vibrant reds and oranges typically depend on carotenoids, accumulated, processed, and eventually deposited in developing feathers (*Lopes et al., 2016*; *Mundy et al., 2016*; *Toews et al., 2017*). In lizards, reds are most often the result of pteridine pigments (*Olsson et al., 2013*), contained within xanthophores, whereas in amphibians and teleosts these colors can result from pteridines as well as carotenoids in xanthophores and erythrophores (*Matsumoto, 1965*; *Wedekind et al., 1998*; *Grether et al., 2001*; *Bagnara and Matsumoto, 2006*; *Sefc et al., 2014*). Our analyses show that in *D. albolineatus*, red and orange colors of adult erythrophores and xanthophores result from carotenoids, detectable by HPLC and lost in *scarb1* mutants. Though pteridines were detectable histologically in erythrophores these did not affect color in the visible range. These observations are concordant with findings from zebrafish, in which the yellow–orange color of xanthophores in adults depends on carotenoids (*Saunders et al., 2019*), whereas yellow coloration of the same cells in embryos and early larvae depends on pteridines (*Ziegler, 2003*; *Lister, 2019*).

The precise biochemical mechanism whereby yellow carotenoids (e.g., zeaxanthin) are converted into ketocarotenoids in animals remains incompletely understood (*Strange, 2016*; *Toews et al., 2017*). In birds, an essential role has been demonstrated for CYP2J19, which is thought to mediate the C4-ketolation of carotenoids (*Lopes et al., 2016*; *Mundy et al., 2016*). Our analyses show that another member of the cyp2 P450 subfamily, *cyp2ae2*, is important for ketocarotenoid accumulation in erythrophores. The finding that two different members of the cyp2 subfamily may have converged on a role in ketocarotenoid formation, suggests that this subgroup of P450 enzymes may be uniquely poised to evolve ketolase activity. Nevertheless, the biochemical function of these enzymes has yet to be demonstrated in vitro, and full ketolase activity may depend on additional factors. In this regard, the markedly reduced abundance of red carotenoids in *bdh1a* mutant erythrophores may provide further clues to the biochemical mechanism of ketocarotenoid production. Indeed, our finding that *bco1* mutants have reduced levels of both red and yellow carotenoids—contrary to the expected activity of this enzyme in carotenoid degradation (*Harrison and Kopec, 2020*) and observations in other systems—suggests biochemical functions and compensatory interactions in this system worthy of further exploration.

The diversification of pigment patterns in teleost has been accompanied by a diversification of pigment cell types, with several distinct classes of iridophores, xanthophores, and leucophores now

recognized in *Danio* fishes alone (*Oshima and Kasai, 2002*; *Hirata et al., 2003*; *Lewis et al., 2019*; *Saunders et al., 2019*; *Gur et al., 2020*). Additional subtypes of pigment cells and even mosaic pigment cells with properties of more than one type have been recognized in more distant teleosts (*Ballowitz, 1913*; *Goodrich et al., 1941*; *Asada, 1978*; *Goda and Fujii, 1995*; *Goda et al., 2011*; *Goda et al., 2013*; *Djurdjevič et al., 2015*; *Salis et al., 2019*; *Parichy, 2021*). In at least one instance cells of one type can transition directly into another type (melanophores→ melanoleucophores) (*Lewis et al., 2019*), whereas in another instance subtypes derived from a common progenitor (stripe and interstripe iridophores) are refractory to interconversion even when challenged to do so experimentally (*Gur et al., 2020*). Our observations of erythrophore and xanthophore origins and fate plasticity suggest a relatively subtle distinction, perhaps limited to the activation or repression of genes essential for the color difference itself. The particular mechanisms that specify these fates or deployment of particular biochemical pathways, and whether additional phenotypes distinguish these cells remain to be elucidated. Such efforts in *D. albolineatus*, and corresponding investigations to uncover genetic bases of erythrophore loss in zebrafish, will be enabled by the identification of cell-type-specific pigments and the development of methods to quantify pigmentary phenotypes in live animals.

# Materials and methods

## Key resources table

| Reagent type (species or resource)* | Designation | Source or reference | Identifiers | Additional information |
|---|---|---|---|---|
| Genetic reagent (*D. albolineatus*) | Tg(*aox5:nucEos*)*vp37albTg* | This paper | | Transgenic line. Maintained in Parichy lab. Described in Materials and methods. |
| Genetic reagent (*D. albolineatus*) | *scarb1vp38ac1* | This paper | | CRISPR-CAS9 knock-out line. Maintained in Parichy lab. Described in Materials and methods, and *Figure 7—figure supplement 1*. |
| Genetic reagent (*D. albolineatus*) | *scarb1vp38ac2* | This paper | | CRISPR-CAS9 knock-out line. Maintained in Parichy lab. Described in Materials and methods, and *Figure 7—figure supplement 1*. |
| Genetic reagent (*D. albolineatus*) | *cyp2ae2vp39ac1* | This paper | | CRISPR-CAS9 knock-out line. Maintained in Parichy lab. Described in Materials and methods, and *Figure 7—figure supplement 1*. |
| Genetic reagent (*D. albolineatus*) | *bdh1avp40ac1* | This paper | | CRISPR-CAS9 knock-out line. Maintained in Parichy lab. Described in Materials and methods, and *Figure 7—figure supplement 1*. |
| Genetic reagent (*D. albolineatus*) | *bco1vp41ac1* | This paper | | CRISPR-CAS9 knock-out line. Maintained in Parichy lab. Described in Materials and methods, and *Figure 7—figure supplement 1*. |
| Genetic reagent (*D. albolineatus*) | *bco1vp41ac2* | This paper | | CRISPR-CAS9 knock-out line. Maintained in Parichy lab. Described in Materials and methods, and *Figure 7—figure supplement 1*. |
| Recombinant DNA reagent | *mitfa:nucEosFP* | This paper | | Maintained in Parichy lab. Described in Materials and methods. |
| Recombinant DNA reagent | *aox5:palmEGFP* | *McMenamin et al., 2014* | | |
| Software, algorithm | JMP Pro 16 | SAS Institute | | |
| Software, algorithm | GraphPad Prism | GraphPad | | |
| Software, algorithm | Fiji | *Schindelin et al., 2012* | | |
| Software, algorithm | Kallisto | *Bray et al., 2016* | | |
| Software, algorithm | DESeq2 | *Love et al., 2014* | | |

* Additional oligonucleotides and CRIPSR/Cas9 reagents provided in *Supplementary file 1*—Table 2 and 5.

## Fish stocks and rearing conditions

*Danio albolineatus* were derived from individuals collected in Thailand by M. McClure in 1995 (**McClure et al., 2006**), provided to the laboratory of S. Johnson, and then maintained in our laboratory from 2000 until the present. Additional species of *Danio* used for assessing erythrophore complements were obtained directly from the field or through the pet trade [*D. aesculapii; D. quagga*, *D. kyathit* (**McCluskey et al., 2021**) *D. nigrofasciatus*, *D. tinwini* (**Spiewak et al., 2018**); *D. kerri; D. choprae; D. margaritatus, D. eythromicron*] and maintained subsequently in the lab or were observed in the field [*D. meghalayensis, D. dangila* (**Engeszer et al., 2007**)]. Fish were reared under standard conditions to maintain *D. rerio* (~28 °C; 14 L:10D) with larvae fed initially marine rotifers, derived from high-density cultures and enriched with Rotimac and Algamac (Reed Mariculture), with older larvae and adults subsequently fed live brine shrimp and a blend of flake foods enriched with dried spirulina. Stocks of mutant or transgenic *D. albolineatus* were: *scarb1$^{vp38ac1}$*, *scarb1$^{vp38ac2}$*, *cyp2ae2$^{vp39ac1}$*, *bdh1$^{vp40ac1}$*, *bdh1$^{vp40ac2}$*, *csf1ra$^{vp4ac1}$*, Tg(*aox5:nucEosFP*)$^{vp43aTg}$.

CRISPR/Cas9 mutagenesis *bdh1a$^{vp40ac1}$* and *cyp2ae2$^{vp39ac1}$* were generated by injecting one-cell stage embryos with 200 pg sgRNAs and 500 pg Cas9 protein (PNA Bio) using standard procedures (**Shah et al., 2015**). *bco1$^{vp41ac1}$*, *bco1$^{vp41ac2}$*, *scarb1$^{vp38ac1}$* and *scarb1$^{vp38ac2}$* mutagenesis as well as *scarb1* targeted clonal labeling were conducted by injecting one-cell stage embryos with approximately 1 nanoliter of 5 µM gRNA:Cas9 RNP complex (IDT). These AltR CRISPR/Cas9 reagents allowed for highly efficient mutagenesis (**Hoshijima et al., 2019**) even in F0 fish, enabling clonal analyses of rare wild-type cells. For production of mutant lines, individual fish were sorted for anal fin phenotypes at juvenile stages and alleles recovered by intercrossing and outcrossing.

Transgenesis *aox5:nucEosFP* and *mitfa:nucEosFP* plasmids were made by assembling 8 kb *aox5* and 5 kb *mitfa* promoters (**Budi et al., 2011; McMenamin et al., 2014**) with nuclear-localizing photo-convertible fluorophore EosFP using the tol2 Gateway Kit (**Kwan et al., 2007**) and were injected at the one-cell stage with tol2 mRNA at 25 pg per embryo (**Suster et al., 2009**) or 6 pg per embryo for clonal analyses.

## Imaging and image processing

Whole fish were euthanized then embedded in 1 % low-melt agarose, and captured on a Nikon D-810 digital single lens reflex camera with MicroNikkor 105 mm macro lens. Anal fin details were imaged using a Zeiss Axio Observer inverted microscope or Zeiss AxioZoom stereomicroscope equipped with Zeiss Axiocam cameras.

Carotenoid autofluorescence was imaged using a Zeiss LSM880 inverted laser confocal microscope in Airyscan SR mode. Laser intensity for red (excitation wavelength 561 nm) and green (excitation wavelength 488 nm) channels were set to be identical. For comparison with brightfield illumination, anal fins were stabilized with a few drops of 1 % low-melt agarose then specimens transferred to a Zeiss AxioObserver inverted microscope with Axiocam camera. Cells along the mid line of 10th inter-fin ray were imaged. Background of bright-field images was corrected to white using software Fiji imageJ (**Schindelin et al., 2012**): Duplicate> Gaussian Blur = 50 > Image Calculator to subtract background> Invert. For larva to juvenile comparisons (**Figure 1F**), 10 cells in 3rd and 4th inter-fin ray in proximal and distal were imaged. For wild-type and mutant comparisons (**Figure 7D**), 20 cells in 10th inter-fin ray in proximal and distal were imaged. Other fluorescent images (e.g. photoconverted images, *aox5*+ clonal labeling) were acquired using a Zeiss AxioObserver inverted microscope equipped with Yokogawa CSU-X1M5000 laser spinning disk and Hamatsu camera.

Images were captured either as single frames or as tiled sets of larger areas that were then stitched computationally using ZEN Blue software, the Autoblend feature of Adobe Photoshop, or manually in Adobe Photoshop. Color balance and display levels were adjusted manually for entire images as needed, with corresponding transformations applied across matched sets of images (e.g. across genotypes). In some instances, gradients of brightness across fields of view (e.g. proximal to distal) were adjusted by applying inverse density gradients in Adobe Photoshop.

## Fate mapping and lineage analysis

Photoconversion was performed on *Tg(aox5:nucEosFP)* or plasmid-injected F0 *mitfa:nucEosFP*, using a Zeiss LSM 800 scanning laser confocal with a 405 nm laser and ZEN blue software. Fish were subsequently reared in tanks shaded from ambient light to prevent spontaneous photoconversion

(*McMenamin et al., 2014*; *Gur et al., 2020*). Brightfield images were taken before photoconversion and fish inspected to ensure that no photoconversion had occurred as a result. Subsequent imaging used fluorescence channels only, except for end-point imaging in fluorescence followed by brightfield. Although pigments autofluoresce in the same channels as nucEosFP, treatment with epinephrine allowed contracted pigment granules to be distinguished unambiguously from nuclei.

For amputation experiments, fins were transected through the middle of the erythrophore-containing region, and imaged subsequently in brightfield (*Figure 5A*) following treatment with epinephrine. Alternatively, *Tg(aox5:nucEosFP)* adult males were exposed under an external Zeiss HXP 120 V compact light source for 15 min until all nucEosFP+ cells in fins had been converted (*Figure 5C*). Fish were anesthetized and anal fins amputated through erythrophore regions, then reared in a shaded tank as above. Unconverted controls reared in the same tank did not show any converted nucEosFP signal when examined concurrently at subsequent time points. Sham control (photoconverted without amputation) of regeneration experiment in *Figure 5C*. Converted nucEosFP signal remained very strong after 41 days. Images were taken 1 day after amputation and at the end point of the experiment. For excisions of middle fin regions (*Figure 5—figure supplement 2*), internal fin ray and inter-fin ray regions were removed and xanthophores close to the wound photoconverted.

## Reverse transcription polymerase chain reaction (RT-PCR) analysis

Adult male anal fins were dissected and dissociated enzymatically with Liberase (Sigma-Aldrich cat. 5401119001, 0.25 mg/mL in dPBS) at 25 °C for 15 min followed by gently pipetting for 5 min. Cell suspensions were then filtered through a 70 µm Nylon cell strainer to obtain a single-cell suspension. Individual cells were then picked manually under Zeiss Axio Observer inverted microscope. Cells were identified by their morphology: red erythrophores, yellow xanthophores, black melanophores and transparent small skin cells. Total RNAs were isolated by RNeasy Protect Mini Kit (Qiagen) and cDNAs synthesized with oligo-dT priming using SuperScript III Cells Direct cDNA Synthesis System (Thermo). Primers pairs were designed to span exon-intron junctions or long introns for assessing genomic contamination, targeting (*Supplementary file 1—Table 5*). Amplifications were performed using Taq polymerase with 35 cycles of 95 °C for 30 s, 56 °C for 30 s, 72 °C for 15 s.

## Pteridine autofluorescence

To assess pteridine content, amputated fins were imaged after exposure to dilute ammonia (pH 10.0), which liberates pteridines from protein carriers resulting in autofluorescence under DAPI illumination.

## RNA-Seq

Adult male *Danio albolineatus* were euthanized, anal fins were dissected and tissue collected from proximal erythrophore or distal xanthophore regions in PBS. RNA was extracted using TRIzol and Direct-zol RNA MiniPrep Kit. mRNA was enriched using NEBNext Poly(A) mRNA Magnetic Isolation Module and sequencing libraries were constructed using NEBNext Ultra RNA Library Prep Kit for Illumina and sequenced on an Illumina Nextseq-500. Reads were aligned to *Danio rerio* reference genome GRCz11 using Kallisto (*Bray et al., 2016*) and analyzed using DESeq2 (*Love et al., 2014*). RNA-seq data are available through GEO (accession ID GSE174713).

## Carotenoid analyses

Proximal (erythrophore containing) and distal (xanthophore containing) portions of the anal fin were dissected from nine individuals of each genotype and like samples were combined in pools of three for pigment extraction. The pooled fin tissue was homogenized with zirconia beads in 1.2 ml of 0.9 % sodium chloride and protein content was quantified a bicinchoninic acid (BCA) assay (23250, Thermo). Carotenoids were extracted from the homogenates by combining 1 ml methanol, 2 ml distilled water, and 2 ml of hexane:*tert*-methyl butyl ether (1:1 vol:vol), collecting and drying the resulting solvent fraction under nitrogen. Each sample was then split and saponified with 0.02 M NaOH or 0.2 M NaOH in methanol at room temperature to maximize the recovery of ketocarotenoids or other xanthophylls, respectively (*Toomey and McGraw, 2007*). The saponified extracts were then injected into an Agilent 1,100 series HPLC fitted with a YMC carotenoid 5.0 µm column (4.6 mm × 250 mm, YMC). Carotenoids were separated with a gradient mobile phase of acetonitrile:methanol:dichloromethane (44:44:12) (vol:vol:vol) through 11 minutes, a ramp up to solvent ratios of 35:35:30 for 11–21 min and isocratic

conditions through 35 minutes. The column was maintained at 30 °C with a mobile phase flow rate of 1.2 ml min$^{-1}$ throughout. The samples were monitored with a photodiode array detector at 400, 445, and 480 nm, and carotenoids were identified and quantified by comparison to authentic standards (a gift of DSM Nutritional Products, Heerlen, The Netherlands).

### Statistical analyses

Analyses of quantitative data were performed in JMP Pro 16 (SAS Institute, Cary NC). Numerical data presented in figures are provided in *Supplementary file 1*.

## Acknowledgements

Supported by NIH R35 GM122471 to DMP and startup funds from the University of Tulsa to MBT. Thanks to Jin Liu and Samantha Sturiale for assistance with screening, stock propagation and imaging, Lauren Saunders and Andy Aman for assistance with RNA-Seq, and Amber Schwindling for fish husbandry.

## Additional information

### Funding

| Funder | Grant reference number | Author |
|---|---|---|
| National Institute of General Medical Sciences | R35 GM122471 | David M Parichy |
| University of Tulsa | start-up funds | Matthew B Toomey |

The funders had no role in study design, data collection and interpretation, or the decision to submit the work for publication.

### Author contributions

Delai Huang, Conceptualization, Formal analysis, Investigation, Methodology, Supervision, Validation, Visualization, Writing - original draft, Writing – review and editing; Victor M Lewis, Matthew B Toomey, Investigation, Writing – review and editing; Tarah N Foster, Investigation; Joseph C Corbo, Supervision, Writing – review and editing; David M Parichy, Conceptualization, Data curation, Formal analysis, Funding acquisition, Investigation, Methodology, Project administration, Supervision, Validation, Visualization, Writing - original draft, Writing – review and editing

### Author ORCIDs

Matthew B Toomey ![ORCID] http://orcid.org/0000-0001-9184-197X
Joseph C Corbo ![ORCID] http://orcid.org/0000-0002-9323-7140
David M Parichy ![ORCID] http://orcid.org/0000-0003-2771-6095

### Ethics

This study was performed in strict accordance with the recommendations in the Guide for the Care and Use of Laboratory Animals of the National Institutes of Health. All of the animals were handled according to approved institutional Animal Care and Use Committee (ACUC) protocol (#4170) of the University of Virginia. Euthanasia was accomplished by overdose of MS222 followed by physical maceration.

### Decision letter and Author response

Decision letter https://doi.org/10.7554/eLife.70253.sa1
Author response https://doi.org/10.7554/eLife.70253.sa2

## Additional files

### Supplementary files

• Supplementary file 1. Tables of RNA-Seq analyses, reagents, and HPLC retention times.

• Transparent reporting form

## Data availability

Numerical data presented in figure panels are provided in Supplementary File 1. RNA-Seq data has been deposited in GEO and is publicly available, accession #GSE174713.

The following dataset was generated:

| Author(s) | Year | Dataset title | Dataset URL | Database and Identifier |
|---|---|---|---|---|
| Huang D | 2021 | Development and genetics of red coloration in the zebrafish relative Danio albolineatus | http://www.ncbi.nlm.nih.gov/geo/query/acc.cgi?acc=GSE174713 | NCBI Gene Expression Omnibus, GSE174713 |

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
