## [Decision Letter]

**Acceptance summary:**

In this beautifully-illustrated paper, the authors present a number of approaches to address the origin and biology of the erythrophore, a red-pigmented cell present in non-mammalian vertebrates including birds, fishes and amphibians. Their data point to a very close relationship between the red erythrophores and yellow xanthophores, consistent with the view that they are similar cell-types differing principally in the details of their pigment biochemistry. The paper will be of interest to those working across many disciplines, including developmental biologists, evolutionary biologists, and those who study the chemistry of pigmentation.

**Decision letter after peer review:**

Thank you for submitting your article "Development and genetics of red coloration in the zebrafish relative *Danio albolineatus*" for consideration by *eLife*. Your article has been reviewed by two peer reviewers, and the evaluation has been overseen by a Reviewing Editor and Richard White as the Senior Editor. The following individual involved in review of your submission has agreed to reveal their identity: Robert Kelsh (Reviewer #1).

Essential revisions:

1) Please attend to the query from Reviewer 1 concerning the identity of the cells used as landmarks in Figure 5A, and include an additional supplementary figure for further clarification if necessary.

2) As requested by Reviewer 2, please provide additional information to clarify the interpretation of the data presented in Figure 3A,B.

3) Please attend to the additional requested changes from both reviewers, to clarify presentation or interpretation of the data, and to correct occasional typos.

*Reviewer #1:*

The authors address an interesting but neglected issue in pigment cell biology, concerning the developmental origin of red erythrophores, especially in relationship to yellow xanthophores, and the genetic basis for their differing pigmentation. Red-yellow colouration in vertebrates usually arises from accumulation of dietary carotenoids, and often has significant behavioural importance, e.g. as an honest signal of individual quality. This and the biochemistry of carotenoid colour variation is nicely covered in the Introduction, providing helpful background to a broad audience.

The authors document the widespread presence of erythrophore in Danio, highlighting the unusual nature of Zebrafish within the genus as lacking them. They then develop some quantitative and objective measures of the xanthophores and erythrophores based upon Hue and Red:Green autofluorescence ratios, allowing clear distinction of the mature cell-types, and note the often binucleate nature of the erythrophores.

The authors then use a variety of tools to assess, with differing degrees of certainty, the lineage relationships of the erythrophores; together these provide a consistent and convincing picture of shared lineage between the two cell-types. This is consistent with the observed gradual shift in properties of proximal cells from xanthophore-like to erythrophore. A more direct test of the conversion of early xanthophores to erythrophores comes from the clonal analysis of aox5:nucEosFP cells (Figure 4). They then use a fin regeneration assay to assess the plasticity of these cells in the mature adult. This is a neat experiment, but I am struggling with the interpretation of Figure 5A: which cells are being used as landmarks to justify the conclusion that the cells shown are clonally-derived form that single cell in the 5 dpa image? It may be that the full series of images could be provided in a supplementary figure and might make this clear, but the current images do not seem convincing to me. The experiment in Figure 5B is convincing, so conclusion seems sound.

The authors then use a transcriptomic comparison to identify candidate genes influencing erythrophore v xanthophore differentiation. They study 3 with mutant phenotypes affecting these cell-types, identifying likely roles of 3 erythrophore genes. Whilst most of this analysis is beautifully presented, I am confused by Figure 7 in which I think panel D and F as described in the legend are inverted.

As is expected form this lab, the manuscript is generally very carefully and clearly written and includes thorough data presentation and statistical analysis. Conclusions drawn are appropriately nuanced, and justified by data presented. The manuscript provides an important first step in understanding the developmental relationship of erythrophores to xanthophores, and a number of genetic resources for the further exploration of this question.

*Reviewer #2:*

In this paper the authors present a number of approaches to address the biology of the erythrophore, a red-pigmented cell present in non-mammalian vertebrates including birds, fishes and amphibians. This cell type is found in many species of the genus Danio but not in the widely-studied zebrafish (*Danio rerio*), therefore the investigators undertake studies in the relative *Danio albolineatus*, which has a prominent population of erythrophores in its proximal anal fin, with xanthophores (yellow-pigmented cells) located more distally on the anal fin. Both cell types use carotenoid derivatives as pigments, and this spatial arrangement provides an opportunity to address hypotheses concerning their ontogenetic relationship. The authors initially provide two lines of evidence that erythrophores and xanthophores arise from a common progenitor: fate-mapping via mosaic labeling of cells with transgenic markers as well as a stable line expressing a photoconvertible FP, and analysis of residual clones of pigmented cells in fish targeted by CRISPR to eliminate the scarb1 gene required for carotenoid uptake. Additionally, these experiments suggest that both cell types arise in the fin via differentiation from an unpigmented progenitor, as opposed to migration of committed or differentiatied cells from the body of the fish. The presence of bipotential progenitors is shown to be restricted to cells in the distal portion of the fin, while examination of regrowth of amputated fins reveals that xanthophores can be regenerated from unpigmented precursors as well as from division and transdifferentiation of erythrophores from the proximal fin "stub". In the last half of the paper, the authors identify a set of differentially-expressed genes between the erythrophore-enriched and xanthophore-enriched regions of the anal fin and subject a number of those from the former to CRISPR mutagenesis, with notable hits in the cyp2ae2 and bdh1a loci diminishing red coloration (and as shown by HPLC, levels of the corresponding ketocarotenoid astaxanthin.) The result with cyp2ae2 is especially intriguing considering the association of a related cyp2 P450 gene with red coloration in avians. The authors' conclusions are well-supported by their use of complementary approaches and clear presentation of a wide range of types of data.

– I think this is a minor issue, but the interpretation of the results in Figure 3 A and B depend on the observed cells being truly clonal. The legend correctly identifies the cell groupings in A as “presumptive” clones, but perhaps there is some other indirect evidence the authors could provide to support this claim? I am thinking along the lines of how many pigment cells were observed in the presumptive clones or how much area they cover, or a panel showing a wider field of view to suggest how rare/isolated these patches are. There were 10 patches in 7 fish analyzed; perhaps there was a number of fish that had no patches at all? In the case of the GFP abelling, as the authors point out elsewhere there are likely to be cells within the observed clones that do not express the marker; it is also mentioned that the transgene stays on once expression is initiated. Is it possible that some cells (from a different clone) could turn the GFP on after the first cells, but be mistaken for progeny from the original clone? Figure 3B clearly doesn’t seem to be a case of this based on the proximity and arrangement of the cells, but is there a way to eliminate this as a possibility? Are the clones observed between d0 and d36? Anyway, the later experiments with photoconversion support the authors interpretation of these mosaic abelling experiments, which is why I think this is a minor issue, but perhaps some additional clarification could be added.

– Second full paragraph on page 10: in line 3, I could be mistaken, but it is my understanding that melanin is transferred to hairs directly from melanocytes residing at the base of the hair follicle rather than via a keratinocyte intermediate. Also, in line 9, strike the word "with" ("can result from with pteridines").

---

## [Author Response]

Essential revisions:1) Please attend to the query from Reviewer 1 concerning the identity of the cells used as landmarks in Figure 5A, and include an additional supplementary figure for further clarification if necessary.2) As requested by Reviewer 2, please provide additional information to clarify the interpretation of the data presented in Figure 3A,B.3) Please attend to the additional requested changes from both reviewers, to clarify presentation or interpretation of the data, and to correct occasional typos.

Each of these changes has been made, details below.

Reviewer #1:The authors address an interesting but neglected issue in pigment cell biology, concerning the developmental origin of red erythrophores, especially in relationship to yellow xanthophores, and the genetic basis for their differing pigmentation. Red-yellow colouration in vertebrates usually arises from accumulation of dietary carotenoids, and often has significant behavioural importance, e.g. as an honest signal of individual quality. This and the biochemistry of carotenoid colour variation is nicely covered in the Introduction, providing helpful background to a broad audience.The authors document the widespread presence of erythrophore in Danio, highlighting the unusual nature of Zebrafish within the genus as lacking them. They then develop some quantitative and objective measures of the xanthophores and erythrophores based upon Hue and Red:Green autofluorescence ratios, allowing clear distinction of the mature cell-types, and note the often binucleate nature of the erythrophores.The authors then use a variety of tools to assess, with differing degrees of certainty, the lineage relationships of the erythrophores; together these provide a consistent and convincing picture of shared lineage between the two cell-types. This is consistent with the observed gradual shift in properties of proximal cells from xanthophore-like to erythrophore. A more direct test of the conversion of early xanthophores to erythrophores comes from the clonal analysis of aox5:nucEosFP cells (Figure 4). They then use a fin regeneration assay to assess the plasticity of these cells in the mature adult. This is a neat experiment, but I am struggling with the interpretation of Figure 5A: which cells are being used as landmarks to justify the conclusion that the cells shown are clonally-derived form that single cell in the 5 dpa image? It may be that the full series of images could be provided in a supplementary figure and might make this clear, but the current images do not seem convincing to me. The experiment in Figure 5B is convincing, so conclusion seems sound.

We added a supplementary figure (Figure 5—figure supplement 1) to show more context and nearby landmarks, including the amputation plane. We additionally swapped out the images in Figure 5A with an example that more clearly makes our point that cells seem to both lose red coloration and increase in number. Cells of both the original and the new example are visible in the new supplemental figure. Given the concern expressed we additionally modified the salient portion of the text, to make it clearer that the brightfield-only analyses were intended merely to see if a transformation is plausible, based on overt cell colors and behaviors in the absence of formal clonal analysis. The revised text reads:

“We first assessed the possibility that transfating occurs by repeatedly imaging individual fish in brightfield, to learn whether cells near the amputation plane might lose their red color during regenerate outgrowth. Individual erythrophores could often be reidentified using other cells as well as distinctive features of fin ray bones and joints as landmarks (Figure 5A; Fig-ure 5—figure supplement 1). As regeneration proceeded, small groups of cells having paler red or orange coloration, were sometimes observable where individual cells of deep red col-oration had been found, suggestive of proliferation and dilution of pre-existing pigments. Later, only yellow cells were found in these same locations. These observations were con-sistent with the possibility of erythrophore → xanthophore conversion, and so to test this idea directly we marked nucEosFP+ erythrophores by photoconversion prior to amputation (Figure 5B; Figure 5—figure supplement 2A).”

The authors then use a transcriptomic comparison to identify candidate genes influencing erythrophore v xanthophore differentiation. They study 3 with mutant phenotypes affecting these cell-types, identifying likely roles of 3 erythrophore genes. Whilst most of this analysis is beautifully presented, I am confused by Figure 7 in which I think panel D and F as described in the legend are inverted.

We fixed the relative ordering of panels and legends. We also changed the Y axis label in Figure 7F to indicate cells per 40 μm^2^ rather than density, which might be misinterpreted to mean cells per mm.

Reviewer #2:[…]– I think this is a minor issue, but the interpretation of the results in Figure 3 A and B depend on the observed cells being truly clonal. The legend correctly identifies the cell groupings in A as "presumptive" clones, but perhaps there is some other indirect evidence the authors could provide to support this claim? I am thinking along the lines of how many pigment cells were observed in the presumptive clones or how much area they cover, or a panel showing a wider field of view to suggest how rare/isolated these patches are. There were 10 patches in 7 fish analyzed; perhaps there was a number of fish that had no patches at all? In the case of the GFP labeling, as the authors point out elsewhere there are likely to be cells within the observed clones that do not express the marker; it is also mentioned that the transgene stays on once expression is initiated. Is it possible that some cells (from a different clone) could turn the GFP on after the first cells, but be mistaken for progeny from the original clone? Figure 3B clearly doesn't seem to be a case of this based on the proximity and arrangement of the cells, but is there a way to eliminate this as a possibility? Are the clones observed between d0 and d36? Anyway, the later experiments with photoconversion support the authors interpretation of these mosaic labeling experiments, which is why I think this is a minor issue, but perhaps some additional clarification could be added.

We now detail in the Methods and Figure 3 the rarity of wild-type scarb1 clones and aox5:palm- EGFP clones, noting that scarb1 mutagenesis used highly efficieny AltR reagents (leaving very few wild-type cells even in F0 fish) whereas aox5:palmEGFP injections were done at limiting diluations. We also added Figure 3—figure supplement 1 to illustrate two more transgene-marked clones. We further identified a typographical error in our reporting of clone compositions in the original manuscript: only 1 clone contained just erythrophores, not 10 as had been written.

Methods: “These AltR CRISPR/Cas9 reagents allowed for highly efficient mutagenesis (Hoshijima et al., 2019) even in F0 fish, enabling clonal analyses of rare wild-type cells.”

Methods: “aox5:nucEosFP and mitfa:nucEosFP plasmids were made by assembling 8 kb aox5 and 5 kb mitfa promoters (Budi et al., 2011; McMenamin et al., 2014) with nuclear-lo-calizing photoconvertible fluorophore EosFP using the tol2 Gateway Kit (Kwan et al., 2007) and were injected at the one-cell stage with tol2 mRNA at 25 pg per embryo (Suster et al., 2009) or 6 pg per embryo for clonal analyses.”

Figure 3A legend: “(8 of 10 presumptive clones in 7 fish, with remaining clones only contain-ing one or the other cell type; an additional 56 fish derived from injected embryos either lacked wild-type cells or lacked mutant cells and were thus uninformative).”

Figure 3B legend: “For these analyses, limiting dilutions of aox5:palmEGFP were injected into ~500 embryos, yielding 271 embryos that exhibited some fluorescence at 3 days post-fertilization that were further sorted at 16 dpf, identifying 27 individuals with patches of ex-pression in the anal fin. Of these 27 fish, one subsequently died and 8 were found to have broad expression across the entire fin, likely representing multiple clones of uncertain boundaries. And so were excluded from analysis. The remaining 18 fish exhibited 24 spatial-ly distinct, presumptive clones of aox5:palmEGFP abelled cells, of which 22 presumptive clones contained both erythrophores and xanthophores as shown here [consistent with mixed clones of melanophores and xanthophores in zebrafish (Tu and Johnson, 2010; Tu and Johnson, 2011)]; 1 clone contained only erythrophores and 1 clone contained only xan-thophores.”

“Figure 3 — figure supplement 1. Transgene labeling of erythrophores and xanthophores. A and B show additional examples of presumptive clones labeled by injection of limiting amounts of aox5:palmEGFP transgene (36 d, 15 mm SL). In each fin, EGFP+ cells can be found across proximal to distal regions of the fin, contributing to erythrophore and xan-thophore populations, respectively. In B, additional EGFP+ cells are evident on the body, at the base of the fin (arrowhead). Asterisks indicate regions of xanthophore autofluorescence in EGFP channel. Images were assembled computationally from tiled acquisitions. Scale bar: 1 mm.”

– Second full paragraph on page 10: in line 3, I could be mistaken, but it is my understanding that melanin is transferred to hairs directly from melanocytes residing at the base of the hair follicle rather than via a keratinocyte intermediate.

We rechecked this issue and it seems that even in hair development melanosomes are received by keratinocytes. We have added citations for this point.

Also, in line 9, strike the word "with" ("can result from with pteridines").

Fixed.